# More Rigorous Software Engineering Would Improve Reproducibility in Machine Learning Research

## Abstract

While experimental reproduction remains a pillar of the scientific method, we observe that the software best practices supporting the reproduction of machine learning (ML) research are often undervalued or overlooked, leading both to poor reproducibility and damage to trust in the ML community. We quantify these concerns by surveying the usage of software best practices in software repositories associated with publications at major ML conferences and journals such as NeurIPS, ICML, ICLR, TMLR, and MLOSS within the last decade. We report the results of this survey that identify areas where software best practices are lacking and areas with potential for growth in the ML community. Finally, we discuss the implications and present concrete recommendations on how we, as a community, can improve reproducibility in ML research.

## 1 Introduction

Scientific claims can only be considered empirical or scientific if we are capable of testing them (Popper, 2005). Theoretical papers are often self-contained, while numerical work typically depends on code. The large numerical part of machine learning (ML) research, therefore, rests upon empirical foundations. Code reproducibility is a key concern in this branch, as it enables others to repeat and build upon prior work.

This paper focuses on what we, as a community, are already doing to enable the reproducibility of our work and how we can improve further. Since machine learning heavily relies on software, we review established software engineering best practices, systematically estimate their adoption in source code repositories associated with publications in major ML conferences and journals between 2018 and 2025, and ultimately argue for more rigorous software engineering in ML research. Our assessment is based on data from a large-scale web crawl of repository links in conference papers. Using the crawled data, we identify current research trends and gaps in software engineering. Given the popularity of ML frameworks like PyTorch (Paszke et al., 2017), Jax (Bradbury et al., 2025), and Tensorflow (Abadi et al., 2015), Python has become the most widely used language in ML. Consequently, the NeurIPS guide for releasing code (Stojnic et al., 2020) is written exclusively with Python in mind. Therefore, while our ideas for examining reproducibility are language-agnostic, our assessment focuses on Python-specific best practices.

Finally, we provide actionable recommendations for how we as a community can improve further. We make the source code for our analysis available online at `https://github.com/BonnBytes/ml-swe-analysis` such that it can be periodically refreshed and later extended to cover additional conferences and journals. Furthermore, the supplementary includes the code in a zip file.

## 2 Related work

Before reviewing software engineering best practices, we survey recent works from the ML community on the hallmarks of reproducibility, how new ideas in reproducibility have been operationalized, the relationship between open source and reproducibility, and the effect of reproducibility on uptake and citation.

**Defining reproducibility**  Tatman et al. (2018) proposed a distinction between ML reproduction and ML replication in which reproduction refers to recreating the exact results reported by a paper, while replication describes the application of the described methods to another dataset.

**Hallmarks of reproducibility**  Hutson (2018) and Haibe-Kains et al. (2020) described how missing source code and data are key obstacles to ML reproduction. However, Tatman et al. (2018) suggested that in addition to source code and data availability, clear communication of software dependencies is another important obstacle towards ML reproduction on machines that are different from the one where a code was originally developed. In addition to code dependencies, we require exact descriptions of algorithmic details. To study these aspects, Raff (2019; 2021) developed a survival model to predict how long it would take to re-implement and reproduce the results from a paper based on features of the paper, like whether it contains pseudocode, a hyperparameter specification, et cetera.

Neighboring computational fields have started to identify code quality as a key component to reproducible research (Hoyt et al., 2023; Ziemann et al., 2023; Pérez-Riverol et al., 2016; Prlic & Procter, 2012; Sandve et al., 2013; List et al., 2017). Similarly, in spirit, this work focuses on best practices for code quality in Section 3.4 and Section 3.5 we discuss best practices like dependency documentation and packaging.

**Operationalization of reproducibility Ideas**  NeurIPS recently published a reproducibility checklist and updated its code submission guidelines to encourage reproducibility (Pineau et al., 2021). In our data, we see a small improvement trend around 2020, when the NeurIPS code guide (Stojnic et al., 2020) went into effect. Section 5 provides a further elaboration. Similarly, ICLR began asking authors in 2022 [1] to include an optional reproducibility statement, while the Machine Learning Reproducibility Challenge (MLRC) encourages investigations into ML reproducibility [2]. In the life sciences, Heil et al. (2021) introduced tangible criteria in the form of a three-tiered reproducibility scale in which the first level requires data, models, and code to be shared along with the paper. The second level requires projects to document software dependencies, the order of commands necessary for reproduction, and to deactivate all stochastic code elements. The third level, or "gold standard", requires enabling reproduction of a paper's analysis with a single command.

**Criticisms of open source**  The ML community does not unanimously argue in favor of open source code releases. For example, Raff & Farris (2023) expressed concerns that releasing open source code could first lead to a relaxation of standards for detailed descriptions within papers and second could enable divergence between code and paper. We suggest that these are instead editorial and peer review issues, which could be alleviated with the improved application of software engineering best practices that both support review and enable automated testing for key assumptions communicated in the paper.

**Reproducibility bolsters citations**  Finally, the statistical analysis by Raff (2023) suggested that reproducible articles are cited more frequently. Following best practices is, therefore, both in the author's and in the community's best interest. The next section describes these best practices.

# 3 Best practices

We review a subset of software engineering best practices and comment on how they can be applied by the ML community. Here, we exclusively consider those that can be automatically measured.

## 3.1 Licensing software

A license communicates the terms under which source code can be used, changed, and distributed. Without a license, source code can not be (legally) reproduced, modified, nor distributed [3]. A permissive license, such as one suggested by the Open Source Initiative [4], enables others to improve, reuse, and extend the

---

[1] `https://iclr.cc/Conferences/2022/AuthorGuide`, reproducibility in not part of the 2021 guide at `https://iclr.cc/Conferences/2021/AuthorGuide`.

[2] Princeton-AI-Lab (2025), `https://reproml.org/`

[3] On GitHub, the terms of service apply, minimally allowing all repositories to be viewed and forked, even without a license

[4] `https://opensource.org/`

code (Sonnenburg et al., 2007; Pérez-Riverol et al., 2016). This can further extend the life of a project since development can continue even without the original authors. Sonnenburg et al. (2007) and `https://choosealicense.com/` offer guidance for choosing an appropriate license.

### 3.2 Onboarding new users with a README

A README file is typically the first documentation that a reader checks in a source code repository. It should include a project description, a guide for installation, a quick start guide, and share information on how to contribute (Stojnic et al., 2020; Pérez-Riverol et al., 2016). Providing a license and README is a language-agnostic step. Below, we focus on the specifics of Python, the most common programming language in ML research.

### 3.3 Formatting, linting, and type checking

Programming language communities often establish style conventions to make source code more uniform and reduce the cognitive burden on readers. The Python community suggested best practices in PEP-8 (van Rossum et al., 2001) and has several tools for automatically formatting code (e.g., Ruff, Black) and for linting code (e.g., Flake8, PyFlakes, Ruff). Optional static type hints (Rossum et al., 2014) enable the implicit documentation of functions as well as the ability to check formal correctness and identify bugs using static type checkers like MyPy (mypy developers, 2025).

### 3.4 Enumerating dependencies

Most ML projects written in Python depend on external Python code, such as PyTorch. Therefore, it is crucial to enumerate these dependencies such that they can be automatically installed. A historical approach has been to enumerate the direct dependencies (i.e., those appearing in the ML code) in a 'requirements.txt' file, which can then be installed via 'pip install -r requirements.txt' (Pip-developers, 2025; Stojnic et al., 2020). The 'requirements.txt' has been historically created by running 'pip freeze', which outputs a lock file that contains all currently installed Python packages (in the current environment) with version pins following manual installation via 'pip install'.

There have been several iterations of project management tools and configuration formats that attempt to systematize the declaration of direct dependencies, including setuptools, Poetry, Hatch, PDM, and ultimately uv. Each tool has historically created its own configuration files (`setup.py`, `poetry.toml`, `hatch.toml`, `pdm.toml`, `uv.toml`), which motivated the Python community to define a standard configuration format and filename `pyproject.toml` in PEP-621 (Cannon et al., 2020). Further, many of these project management tools performed similar locking operations to `pip freeze`, which created their own lock files (`Pipfile.lock`, `poetry.lock`, `uv.lock`, etc.) that can be used to reproduce an exact environment. The Python community defined a standard configuration format and filename `pylock.toml` in Python Enhancement Proposal (PEP)-751 (Cannon, 2024), which will simplify reproduction in the future.

Conda (Conda-developers, 2025) is a more generic package manager that can install both Python and non-Python dependencies, then generate a 'environment.yml' file that lists dependencies. However, Conda is not a packaging tool and is often incorrectly used as a substitute for properly packaging Python code.

### 3.5 Packaging code

Most ML projects re-use functions or even entire modules from others. While directly copying code is simple, it is problematic for maintainability and visibility (Pérez-Riverol et al., 2016). For example, if a bug is identified, it must be manually fixed for every duplication. Packaging solves this problem by enabling code to be uploaded to the Python Package Index (PyPI), installed with package installer for Python (PIP), then imported using standard `import` statements without local code duplication.

Packaging is relatively straightforward: A packaged project should have `src` and `tests` folders (Python-Packaging-Authority, 2025). Metadata about the package, such as its name, license, and dependencies, can be encoded in a standard `pyproject.toml` file (Brett Cannon, 2016; Smith et al., 2015).

The `pyproject.toml` also specifies a *build backend* (e.g., `setuptools`), which produces artifacts for PyPI (Python-Packaging-Authority, 2025; Smith et al., 2015). When authors use `setuptools` (Setuptools-Team, 2025) as a build backend, they also need to include a `setup.py` or `setup.cfg`. Hatch (Hatch-developers, 2025), Poetry, and uv can be used as a setuptools alternative for users seeking a high-level experience with carefully chosen defaults.

By packaging our code, we enable others to replicate and build on our work via automatic installation. The process gives users access to a working installation now and to package updates and bug fixes in the future. Generally, we recommend widespread adoption of the backend-agnostic `pyproject.toml`-file (Brett Cannon, 2016).

## 3.6 Testing

Tests allow us to check new ideas before integrating them into the core codebase (Pérez-Riverol et al., 2016). By writing tests, we document core development assumptions and the key behavior of the code. When other groups or new people in the same group join a project, the tests help them ensure older features remain operational as new functionality is added. Two popular test frameworks for the Python programming language are `pytest` (Pytest-developers, 2025) and `unittests` (Python-developers, 2025). Regarding test organization, the Python Packaging Authority (PyPA) recommends organizing them into a `tests` folder (Python-Packaging-Authority, 2025) in the root directory of the project. Within the Jax ecosystem, Chex (ecosystem authors, 2025) offers an elegant and modular way to automatically evaluate individual tests in different ML-specific settings. For example, its `@variants` decorator automatically tests a function's compatibility with just-in-time (JIT) compilation, device mapping, and parallelization.

Tests allow maintainers, contributors, and users to verify that the code works as intended, which fosters confidence in new contributors. In this setting, successful test runs indicate that everything is set up correctly (List et al., 2017). In the long run, tests ensure that the code remains functional as it evolves. Without it, bugs may be introduced, but not discovered. Such hidden errors are a form of technical debt (Breck et al., 2017; Amershi et al., 2019). During the development phase, automated tests facilitate code verification and prevent the accumulation of technical debt. The next section focuses on the test automation.

### 3.6.1 Automation

After configuring formatting, linting, type checking, packaging, and testing for a project, it is possible to standardize, consolidate, and automate their application using workflow tools like `tox` or `nox`. These tools automate the creation of isolated virtual environments, installation of packaged code, and allow for arbitrary configuration of other build steps that need to be run, e.g., before tests. They are configured with a `tox.toml`/`tox.ini` or `noxfile.py`, respectively, which appear in the root directory of the project.

Finally, Continuous Integration (CI) can be used to run these workflow tools, e.g., on all pushes to a project on GitHub. On GitHub, this can be configured via a yaml file placed in the `.github/workflows` directory. Overall, CI facilitates enhanced cooperation between team members through automatic verification of new features.

### 3.6.2 Recording seeds

Because neural network optimization is typically not a convex problem, we often require pseudorandom initialization to initialize network matrices before we begin experimenting. However, in order to be reproducible, stochastic behavior must be disabled and the seed given to the random number generator must be made explicit (Heil et al., 2021). In this regard, Jax (Bradbury et al., 2025), for example, makes the state of the pseudorandom number generator explicit by introducing a unique key object. Similarly, PyTorch users can optionally choose to set seed values (PyTorch-Contributors, 2024).

### 3.7 Documenting code

Documentation is the component that makes code accessible to others. Typically, documentation is generated in the form of docstrings for every user-facing function, class, and module. Automated tools like Sphinx (Sphinx-developers, 2025) allow us to generate documentation websites from the docstrings. It is customary to create a `docs` folder for the documentation (Sphinx-developers, 2025). Specialized web services such as ReadTheDocs[5] automate the build and hosting of documentation.

### 3.8 Difficulties in adopting best practices

**- updated: »  Johanson & Hasselbring (2018) observed low adoption rates of modern software engineering techniques in computational sciences and found that few scientists are trained in software engineering. Arvanitou et al. (2021) confirmed this observation for scientists funded by the United States National Science Foundation (NSF) and also identified the issue of scientists struggling to schedule time for training in software engineering. While the ML-community is a different group of people, we suspect a similar situation. Furthermore, we argue that professors don't place enough value on software engineering, perhaps because many have no experience working this way. We also argue that funding bodies and journal editor's attitudes matter - Lee et al. (2021) suggests both could look for evidence of software engineering as a quality signal.**

**Additionally, working under tight conference deadlines likely exacerbates this problem. We have found, both through personal experience and through anecdotal evidence, that after a steep initial learning curve, proper software engineering practices are not difficult to follow, especially since we can rely on pre-configured software repository templates[6]. In the long run, especially when considering project handover from one PhD student generation to another, software engineering saves time.  «**

## 4  Methods

We developed an automated pipeline that quantifies the adoption of the software engineering best practices described in the previous section. Our focus is the ML community, where lots of research appears in conference proceedings. We generated criteria for choosing ML journals and conferences based on their generality and reputation. We limited ourselves to four top conference venues and two journals due to time constraints. The NeurIPS, ICML, ICLR, and the International Conference on Artificial Intelligence and Statistics (AISTATS) conference, as well as TMLR and MLOSS, are included in this study.

The pipeline first downloads Portable Document Format (PDF) documents in bulk from select journals and conferences. We wrote custom web scrapers when proceedings websites are available. Whenever no proceedings had been published yet, we relied on the OpenReview-API to download the proceedings. This was the case for ICML 2025 at the time we crawled the data. First, for proceedings pages, we use beautiful-soup (Richardson, 2023) to extract all links to papers by filtering for links ending with `pdf`. Second, we extracted links to source code repositories hosted on GitHub [7]. We use `pdfx` (Hager, 2021) and a small number of custom natural language processing functions for PDF processing. Finally, in each repository, we look for the existence of the following files and their common spelling variants and file extension variants in order to estimate the adoption of software engineering best practices: (`LICENSE`, `COPYING`, `README`, `requirements.txt`, `Pipfile.lock`, `pylock.toml`, `pyproject.toml`, `tox.toml`, `tox.ini`, `setup.py`, `setup.cfg`, `noxfile.py`, `environment.yml`, `uv.lock`, `poetry.lock`, `poetry.toml`, `hatch.toml`, `pixi.lock`, `pixi.toml`, `.pre-commit-config.yaml`, `Makefile`) and folders (`docs`, `test`, `tests`, `.github/workflows`). Test folders sometimes appear inside the `src` folder or in a folder with the same name as the project. For completeness, we check these locations as well.

---

[5]https://about.readthedocs.com/
[6]https://cookiecutter.readthedocs.io/en/2.0.2/README.html
[7]**- updated: »  Adding links from additional sources like 'papers with code' will be important future work.«**

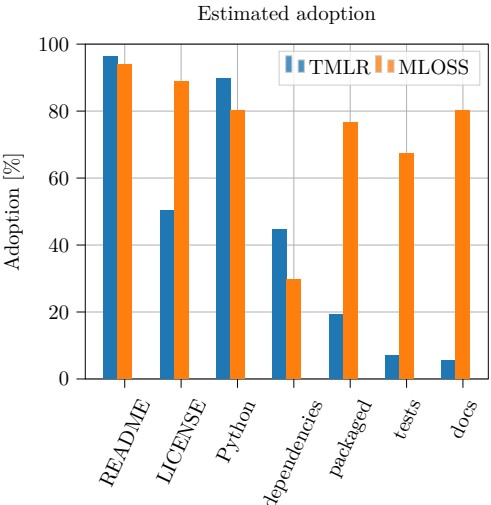

Figure 1: Estimated state of software engineering best practices at TMLR and MLOSS. The software focused MLOSS serves as a baseline for comparison. The plots illustrate web-crawled percentages of files and folders tied to the adoption of software engineering best practices.

**- updated: »  The statistics we are gathering are estimations based on the presences of files and folders named using standard conventions. We argue that this provides a reasonable simplification, since the usage of custom or non-standard configuration does not often correlate with best software practices nor reproducibility, which could lead to minor underestimation. We also assume that files with correct names are used correctly, which could lead to minor overestimation. We observe that in practice, these do not pose a substantial risk to the accuracy of our quantification, as the biggers issue is the presence of such file or folders to start.«**

Before looking at conference proceedings, we first estimate the adoption of software engineering best practices for the journals such as Transactions on Machine Learning Research (TMLR) and Machine Learning Open Source Software (MLOSS) (see Figure 1). We use MLOSS as a baseline because its submissions are generally reusable software, which in turn correlates with the adoption of software engineering best practices. To broaden our view further, we look for signs of best practice adoption within the software repositories we extracted from papers that appeared at major machine learning conferences since 2018.

Since our analysis is Python-specific, the Python adoption rate should be considered as the upper limit when reading Figure 1. To avoid confusion, we exclude repositories that do not use Python later in this section, when we discuss Python-specific methods.

## 5   An assessment of the software ecosystem state in ML research

In Figure 1, we observe that TMLR papers are not yet at the level of MLOSS in terms of best practices adoption. We observe growth potential for TMLR code submissions in almost all dimensions in comparison to MLOSS. The rest of this section considers conference venues where most ML research appears, we will keep the MLOSS-baseline in mind.

**README and LICENSE files over time**   In Figure 2, we observe that the inclusion of README files is widespread, with nearly all repositories across all journals and conferences having full adoption. However, the inclusion of license files seems to have stagnated between 50% and 80% over time, with ICLR, NeurIPS, and ICML having the most. This means users are potentially working without legal security with the implications outlined in section 3.1.

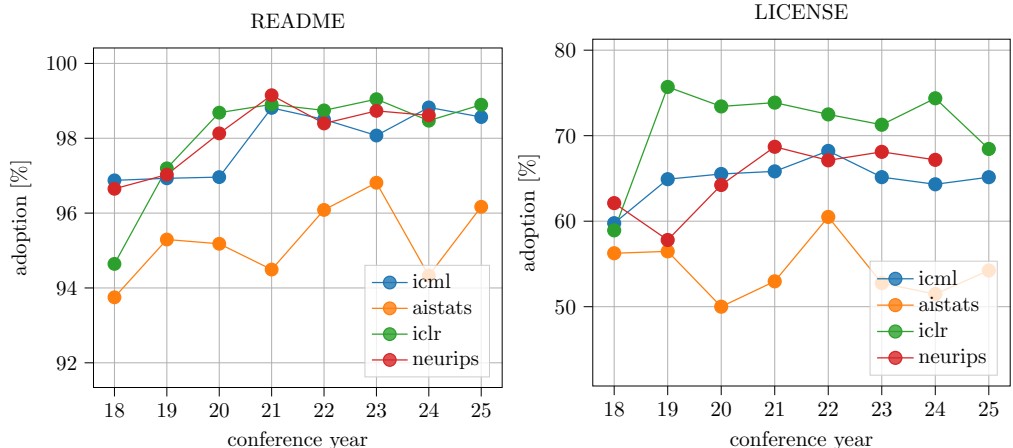

Figure 2: Estimated state of README and LICENSE file adoption in major ML conferences. We add the counts for `README.md` and `README.rst` files, as well as common spelling variations and show these as `README`. For the licenses, we add the counts for `LICENSE`, `COPYING`, and common spelling variations.

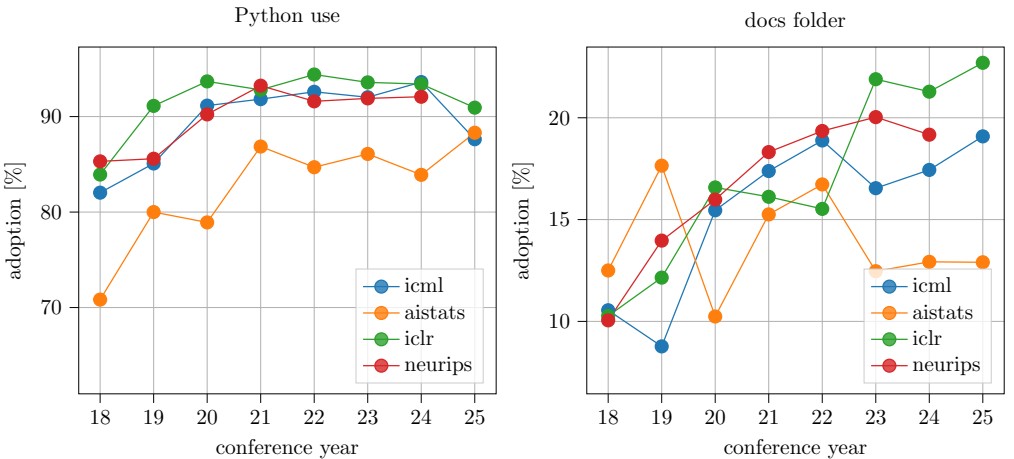

Figure 3: Repository link crawl results for Python adoption (left) and standalone documentation (right). We count each repository where Python is listed as a language in GitHub's language box. The plot on the right of this figure illustrates the share of repositories with a `doc` or `docs` folder.

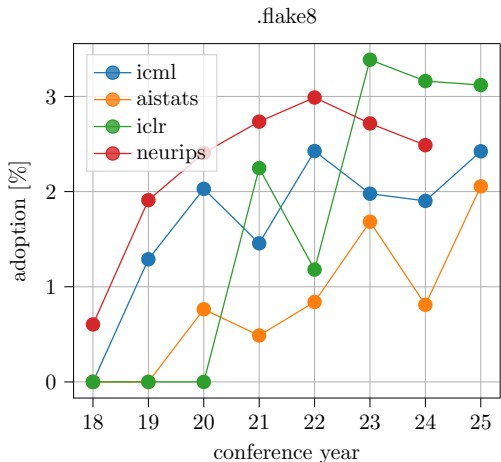

Figure 4: Adoption of `.flake8` configuration files in repositories over time.

**Python adoption over time**   Most of the engineering best practices we described previously in Section 3 are Python specific, before moving on, we must therefore check if Python is indeed the most common language in ML. In the left-hand plot of Figure 3, we observe an upwards trend in the usage of Python over time. Since 2021, more than 80% of repositories use Python at all major conferences we considered. We believe this trend bolsters the case for more rigorous software engineering. Since fewer language barriers exist, packaging especially would help us to collaborate more effectively as a community. After all, since almost everyone is working with Python, the `import` statement is available to the vast majority of the community, and can help us to avoid code duplication.

The following sections will focus on Python-specific engineering practices. We exclude repositories in other languages from our analysis.

**Documentation**   In the right-hand side of Figure 3, we observe that the share of projects with a standalone documentation folder approaches 20%. While this share remains low, we see a clear upward trend, and in some cases, a well-written README file is also sufficient.

**Linting**   In Figure 4, we observe a relatively low, but upwards-trending, adoption of configuration for Flake8. However, tracking `.flake8` is not necessarily representative of linting adoption because of widespread adoption of Ruff[8], which is typically configured in the `pyproject.toml`. Therefore, we can estimate an upper bound on linting by combining `pyproject.toml` and `setup.cfg` file counts from Figure 7 with the `.flake8` numbers from Figure 4, which places us well under 50%.

**Requirements documentation**   Requirements are very important for reproducibility since initializations, for example, differ between PyTorch versions[9]. The exact version used in a project should therefore appear in the requirements documentation. In Figure 5 and Figure 6 depict the numbers. We observe moderate, upwards-trending adoption of various mechanisms for declaring dependencies.

The number of `.lock`-files we found was very small, the `uv` version appeared most (see supplementary Figure 9 for the rare ones). Lockfiles are higher-level application-centric alternatives to the `requirements.txt` file.

Both `requirements.txt` and environment files appear more frequently. At ICML in 2024, the combined share of projects with `requirements.txt` and `environment.yml` files was still less than 50%. This adoption rate implies that the results of many projects will not be reproducible straightforwardly. The NeurIPS code guide appeared (Stojnic et al., 2020) in 2020, it asks authors to provide these files. We see a solid positive trend since the guide appeared. Ideally, we should aim to allow straightforward reproduction for every project.

---

[8]`https://github.com/astral-sh/ruff`

[9]`https://docs.pytorch.org/docs/stable/notes/randomness.html`

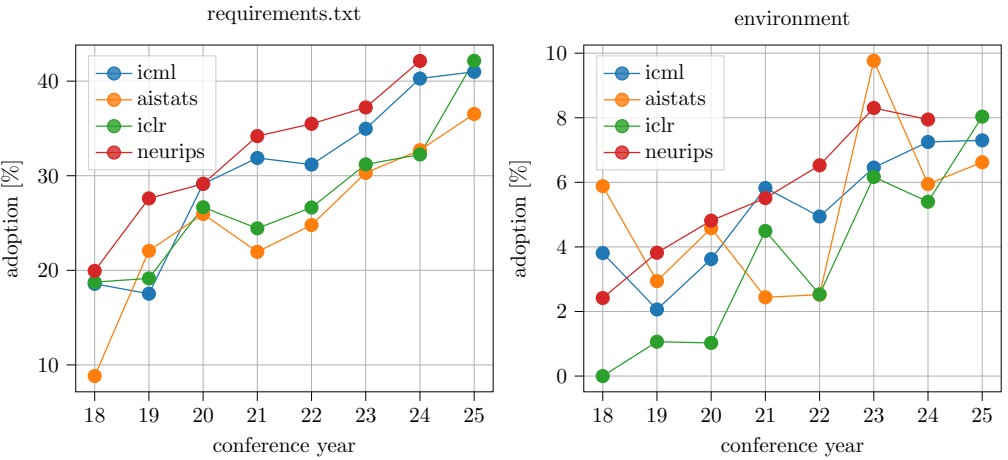

Figure 5: Requirements documentation over time. The figure illustrates the share of repositories with `requirements.txt` and `environment.yml` files.

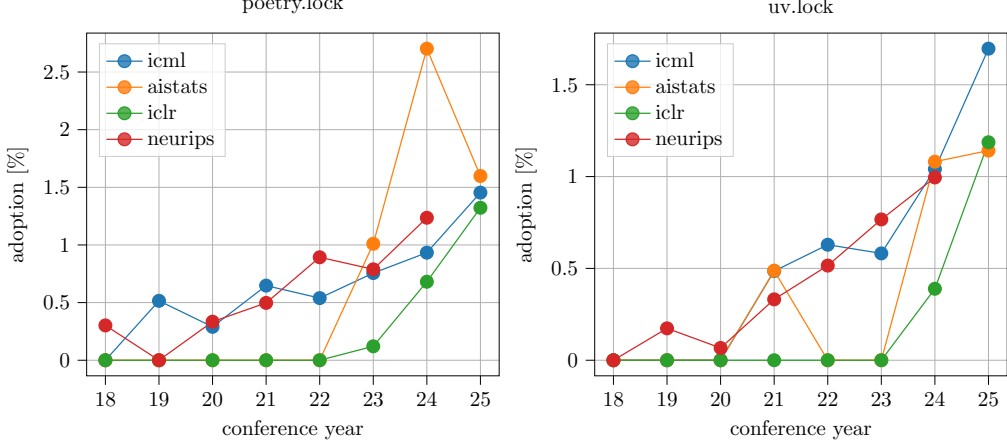

Figure 6: Overview of the lock files the crawler discovered. The environment plot adds the numbers for `environment.yml` as well as `environment.yaml` files.

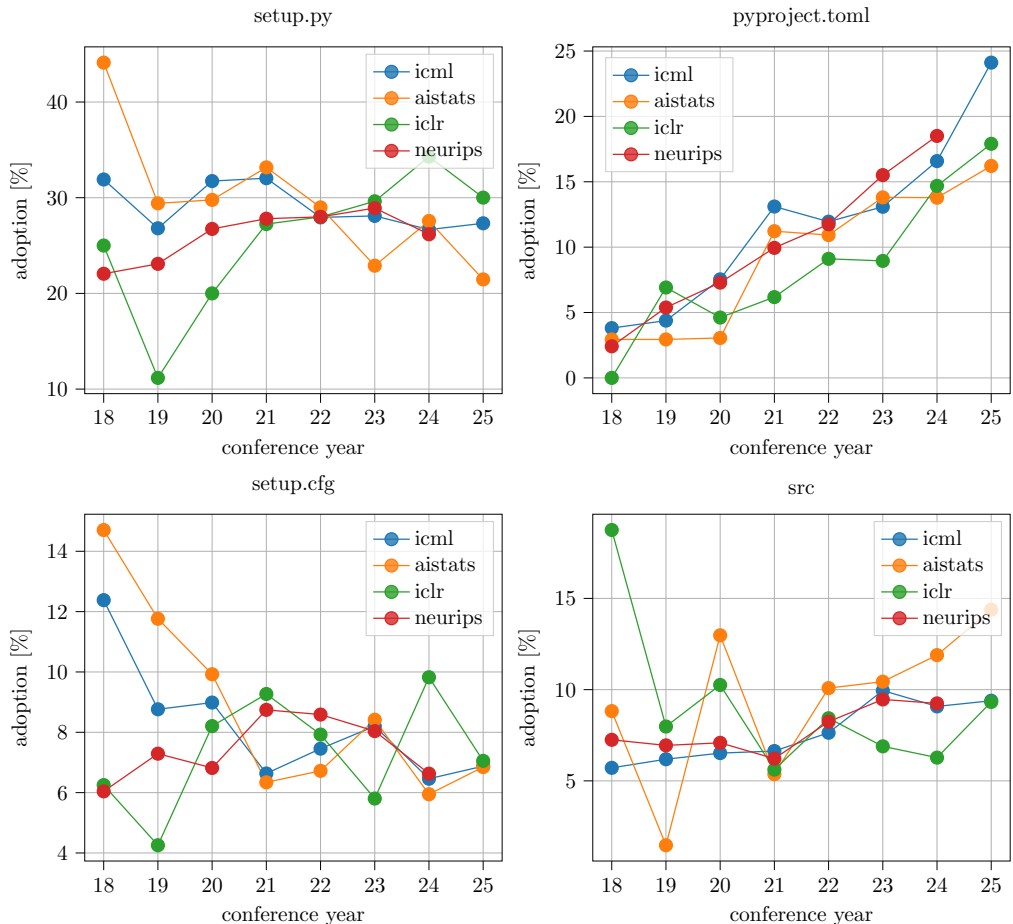

Figure 7: Files and folders which indicate Python packaging adoption over time.

It is possible to provide both files automatically with a single command each. Pip users can run `pip freeze > requirements.txt` to create the historic file. For the modern standard `pip lock -e .` will create the `pylock.toml` file (Cannon, 2024). While we did not find many usages `pylock.toml` due to its recent introduction, we expect to see more of these files in the future. Similarly, for Conda users, providing the file requires typing `conda env export > environment.yml` into the terminal. Afterward, both groups could commit the files to their code repositories, doing so will ensure future scientists end up with the correct software versions.

**Packaging adoption** Packaging as described in section 3.5 is an elegant way to improve code reusability. This section discusses our estimates of Python code packaging at major ML conferences over time. Figure 7 illustrates that the numbers stagnate roughly between 20% and 40% for `setup.py` files. PEP-518 recommends the use of `pyproject.toml` files over `setup.py` files since 2016 (Brett Cannon, 2016). The PEP also outlines the case against `setup.py`. The trend for `pyproject.toml` adoption is upwards, which is encouraging. The `setup.cfg` is a configuration file for setuptools, which is used to package Python code. Its use is falling, presumably because people are migrating to `pyproject.toml` files, as recommended.

We also tracked `hatch.toml` files. In supplementary Figure 10, we see a small upward trend for its use, but it's size is insignificant. Overall, we see that more projects could package their code. This is apparent, especially in comparison to the adoption rates we saw for MLOSS on the right of Figure 1. The current upward trend for `pyproject.toml` files is encouraging. Packaging is a key component of replicable research, and we should encourage authors to package their code, for example, by including a reference to packaging in the NeurIPS code guide (Stojnic et al., 2020).

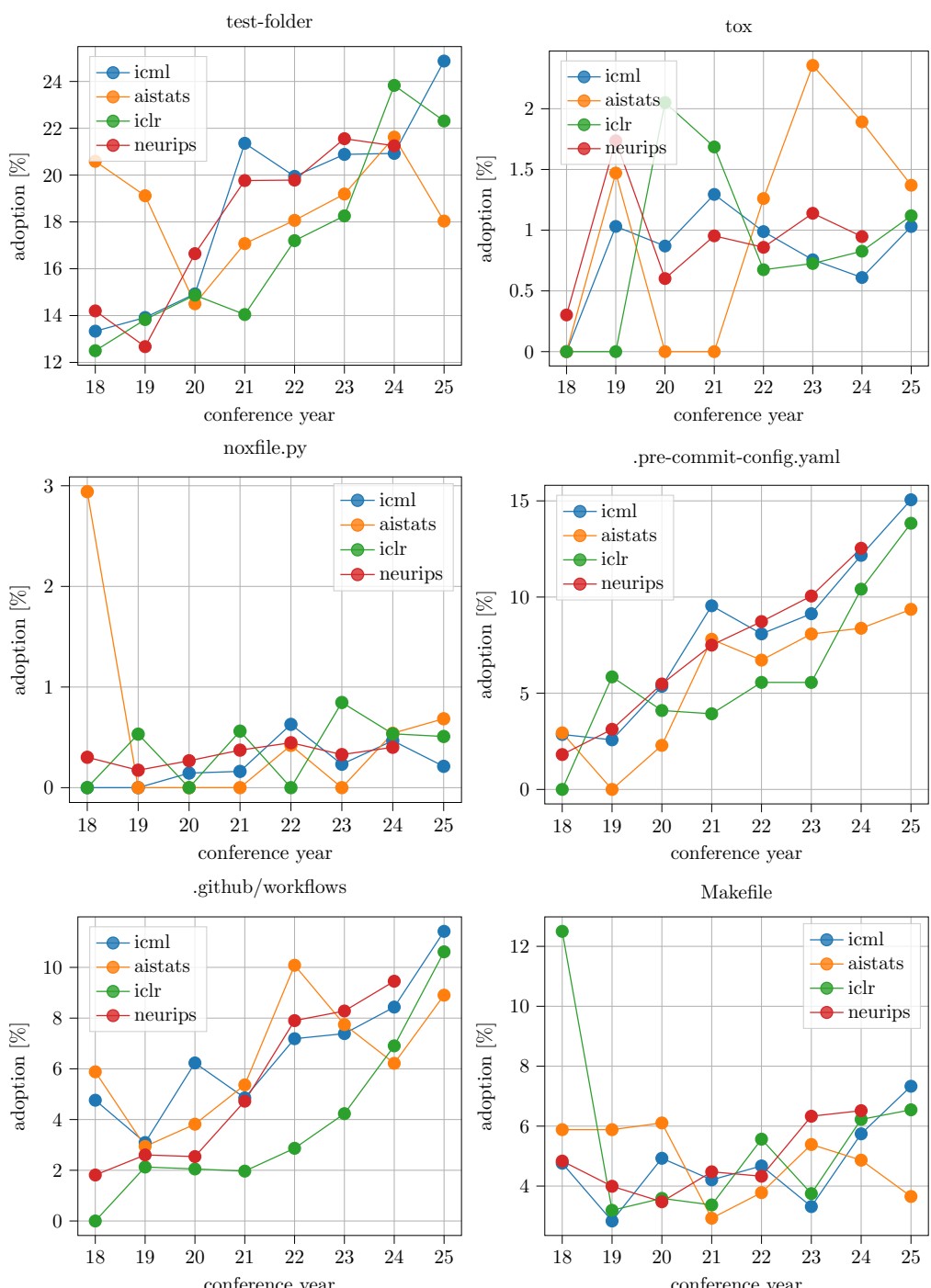

Figure 8: Systematic test adoption over time. The plot for tox combines counts of `tox.ini` and `tox.toml`, both of which are valid filenames.

**Test and workflow use**   Figure 8 illustrates our estimate of the adoption of both methods in parts of the ML community over time. In the top-left plot in Figure 8, we observe the share of repositories where our crawler found test folders. We look at the repository root for `test` or `test` as well as within a `src` or a package folder. We estimate that roughly a quarter of repositories have a dedicated test folder.

Containerized testing requires a specification of dependencies, and a package helps. Consequently, our estimates for both are upper bounds. The numbers from the Tox and `noxfile.py` plots in Figure 8 allow us to estimate the adoption of containerized testing. Since both Tox and Nox run tests in isolated containers. We identified automated workflows by checking the presence of the `.github/workflows` folder and `.pre-commit-config.yaml` files, which are both used to automate workflows, such as by running tests on every push to a repository. Compute-intensive tests are often automated via `.github/workflows`. Linting and formatting checks are less costly and often appear in `.pre-commit-config.yaml` files. Roughly three-quarters of the community work largely without automated testing and workflow automation, even though most projects are team efforts with multiple authors.

## 6 How can we do better?

We believe that a step towards improving the community adoption of software best engineering practices could take the form of a checklist for submitters and reviewers, similar to the checklist proposed by Hoyt et al. (2023) for the cheminformatics community:

- Does the code have a license file?

- Is a README included at the repository root?

- Does the project document its dependencies? Does it provide a `pylock.toml` file? If not, are the dependencies documented in the `pyproject.toml` file?

- Does the project facilitate automatic installation by others via packaging? It's never too late to package code! When deadline pressure prevents us from spending the time early, this can still happen between paper acceptance and conference presentation.

- Is it possible to run tests automatically? Did we test our external code dependencies by running a containerized set of tests?

- Are all pseudorandom number generator seed values fixed?

- Did automatic code checkers like flake8 or MyPy (in case of type annotations) report any problems?

- Ideally, we would like to have a single file or a command that executes the code required to re-run all experiments from a paper.

To improve the current situation, we should ask code authors these or similar questions more frequently when reviewing and read the author's answers with an open mindset.

Sometimes, there are good reasons for a no, which can be an acceptable answer. For example, an illustrative notebook for a theory paper does not require a big test machine, since the paper rests on mathematical proofs. Furthermore, a mindless test crusade risks triggering authors to withhold code altogether. We require a measured approach, which stresses authors' interests and needs. Most of the time, authors will re-use their code themselves. Moreover, proper software engineering will boost their reach. Both arguments can help us to convince authors to adopt best practices.

Additionally, as a reviewer, if we notice that engineering best practices are not observed, we should share relevant sources from the Python community with the authors, for example, by pointing them to the NeurIPS code guide (Stojnic et al., 2020) or to this paper.

## 7 Conclusion

In ML research projects build on each other and teams change over time, motivating software engineering best practices including testing, packaging, and documenting dependencies. Their application saves time and effort and also reaps benefits in the long run by preventing bugs, easing on-boarding, and promoting

reproducibility. Therefore, we advocate for more rigorous software engineering, but with author's interests in mind. However, we recognize that there are practical difficulties with applying best practices, including the lack of good incentives, lack of mentorship, competitiveness of the field, and often time pressure. Therefore, we are mindful when recommending enforcing software engineering practices that they should be applied only when appropriate, e.g., limiting ourselves to cases where systematic tests are efficient and valuable. Forgoing testing, dependency documentation, and packaging initially saves time, which produces a competitive advantage since skipping systematic testing will allow projects to be finished quicker and papers to appear earlier. However, our methodology should not be governed by short-term interests. As large parts of ML research rely on shared code, improving software quality means building a stronger foundation for our work, which is in everyone's best interest.

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

# A   Appendix

## Impact Statement

This work exhibits a critical gap in the adoption of software practices to improve reproducibility in ML research and proposes actionable recommendations. Our findings aim to call for action in the ML community to promote reproducibility and long-term scientific integrity through these standard software practices.

## Acronyms

**AISTATS** International Conference on Artificial Intelligence and Statistics

**CI** Continuous Integration

**ICLR** International Conference on Learning Representations

**ICML** International Conference on Machine Learning

**JIT** just-in-time

**ML** machine learning

**MLOSS** Machine Learning Open Source Software

**MLRC** Machine Learning Reproducibility Challenge

**NeurIPS** Conference on Neural Information Processing Systems

**NSF** National Science Foundation

**PDF** Portable Document Format

**PEP** Python Enhancement Proposal

**PIP** package installer for Python

**PyPA** Python Packaging Authority

**PyPI** Python Package Index

**TMLR** Transactions on Machine Learning Research

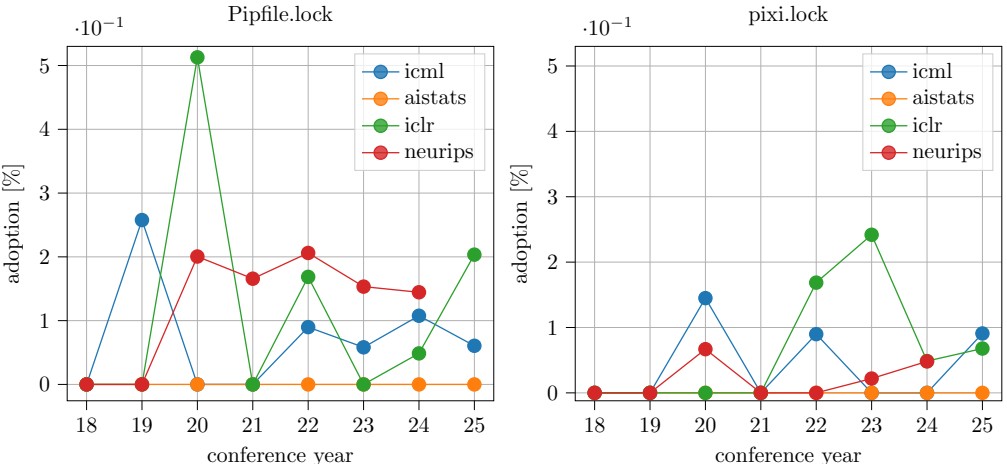

Figure 9: Rare requirements documentation `lock`-files

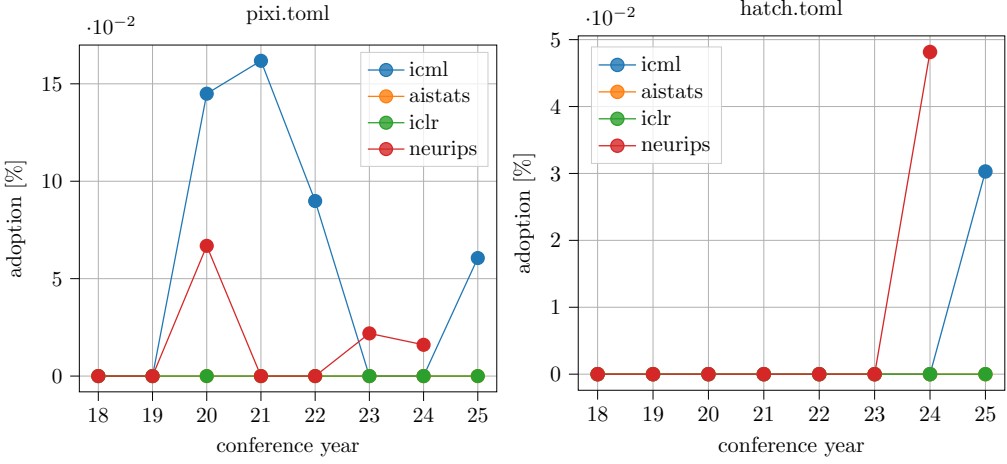

Figure 10: Adoption of potentially emerging packaging tools.

