# OpenReview forum: "More Rigorous Software Engineering Would Improve Reproducibility in Machine Learning Research"
_TMLR — Rejected by TMLR_

### Review · Reviewer_8TzE · 2025-09-30

**Summary Of Contributions:**

The paper builds an automated pipeline that crawls NeurIPS, ICML, ICLR, AISTATS, TMLR, and MLOSS proceedings from 2018 to 2025, extracts GitHub repositories, and audits the presence of files and folders that signal software engineering practice in ML code. It focuses on Python because it dominates ML repositories and enables standardized checks. It measures adoption of licenses, dependency documentation, packaging, testing, automation, and documentation using file‑presence heuristics, then reports venue‑specific and temporal trends. The analysis shows that most repositories lack many of these elements across venues. It contrasts TMLR with the software‑focused MLOSS baseline to contextualize gaps, releases code and a refreshable crawler for replication and extension, and proposes a concise reviewer and author checklist aimed at simple and reliable reproduction, ideally with a single command.

**Strengths**

* The paper targets a core reproducibility bottleneck in ML and defines a concrete measurement target.
* It specifies a clear set of best practices and explains why each matters. The practices cover licensing, README, dependencies, packaging, testing, automation, seeds, and documentation.
* It implements an automated pipeline that crawls proceedings, extracts GitHub links, and audits repositories for file and folder signals. It spans six prominent venues from 2018 to 2025. The scope enables venue comparisons and trend analysis.
* The figures on pages 6 to 10 make the gaps clear and are easy to interpret.
* It provides a short checklist that reviewers and authors can adopt immediately.

**Weaknesses**

* The study uses file presence proxies and does not inspect file contents. This can overestimate real adoption and ignores quality. For example, just because the file "setup.py" is not included in a Github repository does not imply that there is no form of setup script included. In addition, even if a file called "setup.py" is included, does not imply that it is actually used in a meaningful way.
* Repo discovery from PDFs is limited. Many papers do not provide a link to the github repository directly in the paper.

**Audience:**

Yes

**Audience Explanation:**

Reproducibility and code quality are central to TMLR’s readers. This paper examines how often basic engineering practices appear in ML repositories and finds broad gaps. That matters to authors who ship code, to reviewers who must judge it, and to users who try to run it. The checklist and concrete steps give immediate value to researchers across subfields. Editors and meta researchers gain a baseline that can inform policy and tooling.

**Broader Impact Concerns:**

None.

**Claims And Evidence:**

Yes

**Claims Explanation:**

The paper supports its claims with a clear and replicable methodology that crawls six major ML venues from 2018 to 2025, extracts GitHub repositories, and audits predefined signals such as licenses, dependency files, packaging configuration, tests, continuous integration, and documentation using objective file and folder checks. The auditing criteria are specified in advance and applied uniformly across venues and years, and the figures tie each conclusion to the corresponding signal. The methods describe scraping, repository matching, and heuristics in enough detail to see how every number was produced. The authors also state the boundaries of the study, including the Python and GitHub focus and the proxy nature of file‑presence metrics, and they avoid making claims that the data cannot support. Within that scope the evidence is accurate, convincing, and clear about the central finding that most ML papers lack several basic engineering practices.

**Requested Changes:**

Refer to the two weaknesses above. I believe a discussion about the limitations of the crawler would improve the paper. In addition, the paper would be more interesting/trustworthy if the crawler was constructed in a more complex way, that somehow takes the listed weaknesses listed above into account. For example, some kind of AI-driven crawler that can more intelligently detect if a repository consists of the listed components (i.e. tests, packaging, requirements etc). This may however be more relevant for a future work.

---

> ### Author Response · Authors · 2025-10-14
> **Response to 8TzE**
>
> Thank you for your thoughtful review and recognizing the potential impact of
> this work. We appreciate your feedback and respond below.
>
> > The study uses file presence proxies and does not inspect file contents. This
> > can overestimate real adoption and ignores quality.
>
> We agree that our study's focus on file presence rather than file contents may
> result in an overestimation of the usage and/or application of best software
> engineering practices in the machine learning community. However, given that
> this is, to the best of our knowledge, the first large-scale survey of its kind,
> we felt that this was a reasonable simplification. Our results already point to
> limited adoption of best software engineering practices in many venues, even
> with potential overestimation, which further strengthens our call to improve the
> adoption of best software engineering practices.
>
> > For example, just because the file "setup.py" is not included in a GitHub
> > repository does not imply that there is no form of setup script included. In
> > addition, even if a file called "setup.py" is included, does not imply that it
> > is actually used in a meaningful way.
>
> We agree that `setup.py` is not sufficient. Our workflow therefore covers a wide
> variety of setup configuration files, including `setup.cfg`, `pyproject.toml`,
> `uv.toml`, `hatch.toml`, `pixi.toml`, and others. If any of these are available,
> there is a high probability that the project is using the packaging tools that
> are typically configured therein.
>
> We recognize that it's possible for a `setup.py` to be present but not for
> packaging purposes, but we observe that this is very uncommon in practice.
> However, this does further motivate a future investigation of file contents
> could provide more accurate results, such as checking for imports of
> `setuptools` or `distutils` within a `setup.py file`, checking of correct
> configuration inside the various `*.toml` files, etc.
>
> > Repo discovery from PDFs is limited. Many papers do not provide a link to the
> > GitHub repository directly in the paper.
>
> We acknowledge that discovery from PDFs can present challenges for recall.
> However, findability is a fundamental first step towards being able to review,
> learn from, reuse, or adapt code. Papers that do not link code are not
> supporting these crucial steps, so we do not think that it is unreasonable to
> consider papers that do not link to code as not following best software
> engineering practices. We have updated our manuscript with this point
> accordingly.
>
> This being said, there are other text and natural language processing challenges
> that may reduce recall. Therefore, we have updated our discussion to allude to
> the possibility of incorporating additional sources, such as "paperswithcode" to
> supplement PDF-based association between manuscripts and repositories.
>
> > I believe a discussion about the limitations of the crawler would improve the
> > paper. In addition, the paper would be more interesting/trustworthy if the
> > crawler was constructed in a more complex way, that somehow takes the listed
> > weaknesses listed above into account. For example, some kind of AI-driven
> > crawler that can more intelligently detect if a repository consists of the
> > listed components (e.g., tests, packaging, requirements). This may however be
> > more relevant for a future work.
>
> We have updated section four and now clearly state the limitations that come
> with using file presence as a heuristic. At this point we are able to present a
> baseline that comes with a relatively straightforward code implementation, it is
> resource efficient and extensible. As stated previously, we look for adherence
> to the standardized best practices which are listed in the paper. An AI-driven
> crawler would perhaps also detect a custom build system or other non-standard
> components, but we argue that this is not desirable in this case, because we as
> a community cannot manage numerous non-standard builds. As our code will be made
> open-source and is extensible, future work could leverage AI to implement more
> detailed checks of file contents.

---

### Review · Reviewer_wwoJ · 2025-09-30

**Summary Of Contributions:**

This paper analyzes the adoption of software engineering best practices in ML research repositories linked to major conferences and journals. Using a large-scale crawl, the authors track trends in README files, licenses, dependency management, packaging, testing, and CI. They find widespread use of READMEs but much lower adoption of licensing, testing, and packaging, which hinders reproducibility. The paper closes with practical recommendations, including a reproducibility checklist.

Strengths:

1. Timely and relevant topic: reproducibility remains a core challenge in ML.
2. Rigorous large-scale data collection with a systematic pipeline.
3. Actionable recommendations tailored to ML researchers.
4. Effective contextualization with related work and community initiatives.


Weaknesses:

1. Analysis is descriptive but not deeply statistical. Causal insights are limited.
2. Heavy reliance on file presence heuristics may under/overestimate adoption.
3. Limited consideration of social/structural barriers (e.g., incentives, mentorship).
4. Some sections repeat existing community guidelines rather than introducing novel methodology.

**Additional Comments:**

The paper is timely, well-written, and relevant. Some sections repeat community guidelines and could be shortened. More discussion of structural barriers (e.g., incentives, mentorship) would help. Overall, the released dataset and checklist will be useful for the community.

**Audience:**

Yes

**Audience Explanation:**

Reproducibility is central to ML, and TMLR’s audience will value an empirical study of current practices alongside concrete recommendations.

**Broader Impact Concerns:**

The work highlights positive ethical impacts, improving scientific integrity and trust in ML research. The main risk is that overly strict enforcement of best practices could discourage code sharing, which the authors acknowledge themselves.

**Claims And Evidence:**

Yes

**Claims Explanation:**

Yes. The evidence is systematic and transparent, although it is largely based on file presence as a proxy for practice adoption. While not perfect, the result is a reasonable and convincing approach at scale. The findings are consistent with community experience.

The methodology clearly demonstrates gaps in current practices and shows encouraging trends where guidelines have been introduced (e.g., NeurIPS). The claims are cautious and well-grounded, though deeper statistical analysis or subfield comparisons could have added further weight.

**Requested Changes:**

1. Clarify limitations of using file presence as a proxy for best practice.
2. Condense sections that restate existing guidelines.
3. Expand discussion of structural and cultural barriers to adoption.

---

> ### Author Response · Authors · 2025-10-14
> **Response to wwoJ**
>
> Thank you for your review, noticing our large-scale data collection efforts and
> the possible overall impact of our paper. Below we address to the points
> raised in the review.
>
> ### Addressing Stated Weaknesses
>
> > Analysis is descriptive but not deeply statistical. Causal insights are
> > limited.
>
> To the best of our knowledge, we present the first large-scale quantification of
> software engineering artifacts. We argue that such a quantification is an
> important first step towards improving community behavior and towards adoption
> of best software engineering practices. We agree that such quantification does
> not necessarily give insight to root causes, so we have inclided references in
> Section 3.8 to work from Johanson & Hasselbring (2018) and Arvanitou _et al._
> (2021) who concurrently suggested the lack of training in software engineering
> as an important issue. Furthermore, we have included a reference to Lee _et al._
> (2021), who discussed the impact of incentives set by publishers and funding
> bodies. Further causal investigation will be important future work.
>
> > Heavy reliance on file presence heuristics may under/overestimate adoption
>
> We agree that statistics based on file presence may lead to underestimation in
> the presence of custom file names or overestimation due to incorrect usage.
> However, we argue that identifying non-standard usages is antithetical to our
> aims, which are to quantify the adoption of community standards.
>
> Further, standard project management tools like `pip` either would not
> understand non-standard usages, or they would be more difficult for reuse and
> extension. Given that most projects have many dependencies, dealing with many
> different non-standard build specifications would be an ineffective use of
> community time.
>
> > Limited consideration of social/structural barriers (e.g., incentives,
> > mentorship)
>
> To address this point, we have expanded Section 3.8 by drawing upon findings
> from the numerical computing literature. Specifically, we added insights from
> Arvanitou _et al._ (2021), who report that about half of postdocs funded by NSF
> working with scientific software had not received any software development
> training. Additionally, in the large pool of NSF-funded researchers, 75% of them
> reported no time for training.
>
> We also cite Lee _et al._ (2021), who highlighted the potential impacts of
> editorial and funding body policies. They suggested that there do not currently
> exist enough incentives to implement more rigorous software engineering in
> science, since most policies do not consider proper software engineering as a
> quality signal. We suspect this also applies to the machine learning community.
>
> Overall, we would like to limit ourselves to citing previous work on the
> social/structural barriers, since our work's focus is to estimate the current
> software engineering best practice adoption rates in the machine learning
> community. We try to make no claims regarding the underlying reasons for its low
> adoption outside of references to previous work. We agree that this is an
> important point which we hope will come up in future work.
>
> > Some sections repeat existing community guidelines rather than introducing
> > novel methodology
>
> Our goal was not to introduce novel methodology for software engineering best
> practices, but rather to operationalize existing guidelines (e.g., from NeurIPS)
> and make a quantification of their adoption. Importantly, we make a detailed
> summary of these community guidelines to introduce them to the target community,
> who, based on our observations, might not be aware of them.
>
> ### Addressing suggested improvements
>
> > Clarify limitations of using file presence as a proxy for best practice.
>
> We have updated Section 4 accordingly to more clearly argue that file and folder
> presence is a reasonable upper bound, based our comments addressing the issue
> above.
>
> > Condense sections that restate existing guidelines.
>
> We believe that detailed background information is key towards making our work
> as widely accessible as possible. Readers who are already aware of these
> guidelines can gloss over these sections, while readers who are not already
> aware will be able to gain valuable insight. For example, reviewer ajPy was not
> familiar with these guidelines and explicitly requested in their review for this
> section to be extended and additional detail added. Therefore, we have updated
> this section for clarity, but have not condensed it.
>
> > Expand discussion of structural and cultural barriers to adoption.
>
> We have updated section 3.8 by adding the additional literature discussed above.

---

### Review · Reviewer_ajPy · 2025-10-08

**Summary Of Contributions:**

The paper develops an automated pipeline to assess the adoption of best practices in software engineering in papers published in ML conferences/journals. A set of best practices are reviewed and quantitative results on adoption in various venues are provided.

**Audience:**

No

**Audience Explanation:**

In principle everyone is concerned with bad reproducibility of papers published in ML venues. while the paper shows some preliminary results based on best practices that it reviews they won't be sufficient to attract attention and encourage community to follow the guidelines. The issue of bad reproducibility is not only because of bad software engineering practices but also from discrepancies in what is theoretically proposed and what is implemented.

**Claims And Evidence:**

No

**Claims Explanation:**

I am not really convinced that the set of best practices as explained in Section 3 are sufficient to judge reproducibility as claimed in the paper. As I am not an expert in software engineering, I miss concrete arguments about sufficiency of these practices in improving reproducibility.

**Requested Changes:**

While I overall like the idea of the paper, it still seems a preliminary version. All the best practices that the paper talks about are recognised by the community but we have not yet found the way to increase adoption. One of my suggestions would be to actually build a framework for uploading code which can do automatic checks for these best practices. ML venues can then adopt such a framework asking authors to upload their code directly there. While I acknowledge that it is difficult to build such a framework/pipeline, the paper from my point of view does not otherwise add new value to the discussion that is already going on.

---

> ### Author Response · Authors · 2025-10-14
> **Response to ajPy**
>
> We thank ajPy for the constructive review and address their points below:
>
> > I am not really convinced that the set of best practices as explained in
> > Section 3 are sufficient to judge reproducibility as claimed in the paper. As
> > I am not an expert in software engineering, I miss concrete arguments about
> > sufficiency of these practices in improving reproducibility.
>
> The section reviews widely accepted best practices to be followed when
> developing software. These community guidelines are authored by reputable
> sources like the Python Software Foundation (PSF), the Python Packaging
> Authority (PyPA), and the Neural Information Processing Systems (NeurIPS)
> Foundation. Our claim that these measures are indeed sufficient rests on the
> authority of these external sources.
>
> Our paper aims to estimate the state of adoption of the best practices outlined
> in these (and other) sources in today's machine learning software ecosystem.
> Section 3 is meant to illustrate which methods exist, why they are important,
> and which files are used.
>
> > The issue of bad reproducibility is not only because of bad software
> > engineering practices but also from discrepancies in what is theoretically
> > proposed and what is implemented.
>
> We agree that the divergence between the implementation and what is reported in
> the paper can be an issue, however, we consider assessing and addressing this to
> be out of scope for this work. However, we do note that adopting improved
> software engineering practices can make it easier for reviewers to identify
> these issues.
>
> > One of my suggestions would be to actually build a framework for uploading
> > code which can do automatic checks for these best practices.
>
> The code associated with this paper can be repurposed to accomplish this, since
> the relevant functionality that we applied at scale for several venues can be
> also applied to a single article at a time. However, we would caution against
> its immediate large-scale adoption, due to the social aspects of the issue. We
> do not want to risk triggering authors to withhold code altogether. For
> theoretically grounded papers, an additionally submitted Jupyter notebook does
> not require a big test machine. Experimental work, however, gains credibility
> when carefully tested.
>
> > we have not yet found the way to increase adoption
>
> We would like to clarify: This paper aims to present an overview of existing
> established best practices and to estimate their adoption. We see room for
> growth, but we do not propose measures to increase adoption. Doing so would
> require us to collect evidence regarding such measures, which remains out of
> scope for this paper. Consequently, to address this concern, we cite potential
> measures proposed elsewhere, but this is not the main focus of this work.
> Changing community standards is a slow process, which we hope our contribution
> pushes forwards a little bit.

---

### Author Response · Authors · 2025-10-14
**Thank you for reviewing our work**

We appreciate our reviewers' constructive comments and valuable feedback. We are
delighted to read that reviewers liked the overall idea (ajPy), found our work
timely and relevant (wwoj), addressing a topic central to TMLR's readers (8TzE),
that concerns everyone (ajPy). The reviewers appreciated our large-scale
automated data collection efforts (wwoJ, 8TzE), while noting that the evidence
is systematic and transparent (wwoJ), with an analysis that highlights gaps and
is easy to interpret (8TzE), enabling venue comparisons and trend analysis
(8TzE).

Naturally, our reviewers also had questions. The most important line of
questions focuses on the file presence heuristic (wwoJ, 8TzE). To address this
issue, we modified the paper by adding a paragraph in Section 4. We now argue
that identifying non-standard usages is antithetical to our aims, which are to
quantify the adoption of community standards.

---

### Decision · Action_Editor_8XAJ · 2025-12-19

**Recommendation:** Reject

**Audience:**

Yes

**Audience Explanation:**

Yes, software engineering best practices are an important topic in machine learning, and members of the TMLR audience would find the paper’s findings relevant and of interest.

**Claims And Evidence:**

No

**Claims Explanation:**

The main issue with the paper is that, in its current framing and formulation, there is ambiguity about the claims being made, which makes it difficult to assess whether they are supported by the evidence presented, as also noted and discussed with reviewer ajPy.

In particular, the paper can be read as making the claim that the adoption of software engineering best practices implies improved reproducibility in machine learning research, an interpretation that is reinforced by the title (“More Rigorous Software Engineering Would Improve Reproducibility in Machine Learning Research”). However, the empirical evidence provided in the paper primarily concerns estimating the current level of adoption of selected software engineering practices, as the authors themselves note in their response to reviewer ajPy.

This ambiguity and mismatch between the implied claims and the actual scope of the empirical evidence make it difficult to conclude that all claims are supported by accurate, convincing, and clear evidence. I therefore recommend that the authors substantially reframe this aspect of the paper. Once this issue has been addressed, they are welcome to resubmit the manuscript to TMLR.

**Resubmission Of Major Revision:**

The authors may consider submitting a major revision at a later time.